

# Temporal and spatial variations in dust activity in Australia based on remote sensing and reanalysis data sets

Yahui Che[1], Bofu Yu[1], Katherine Bracco[1]

[1]School of Engineering and Built Environment, Griffith University, Brisbane, 4111, Australia

*Correspondence to*: Bofu Yu (b.yu@griffith.edu.au)

**Abstract.** Spatial and temporal variations in the level of dust activity can provide valuable information for policy making and climate research. Recently, MODIS aerosol products have been successfully used for retrieving dust aerosol optical depth (DAOD), especially over bright dust source areas and MERRA-2 aerosol reanalysis provides DAOD, and additionally other dust aerosol-related parameters. In this study, spatial and temporal variations in dust activity in Australia were analyzed using MODIS and MERRA-2 combined (M&M) DAOD and MERRA-2 near-surface dust concentrations/estimated PM10 for the period from 1980-2020. Validation results show that M&M DAOD has an expected error of $\pm(0.016 + 0.15\tau)$ compared to the ground observations at the AERONET sites. MERRA-2 near-surface dust concentrations show a power law relationship with visibility data collected at meteorological stations with an $r^2$ value from 0.18 to 0.44, and the estimated MERRA-2 PM10 shows similar temporal variations and correlates with ground-based PM10 data with an $r^2$ value from 0.14 to 0.44 at six selected stations in Australia. Moreover, MERRA-2 dust flux shows the same major dust pathways as those in previous studies and similar dust emissions/deposition areas. Dust events based on DAOD over eastern Australia are concentrated in the north in December, in the south in February, and can occur anywhere in January. Near-surface dust concentration was found to be the highest (over 200μg/m$^3$) over the center of Lake Eyre Basin in central Australia and radially decreased to the coast to below 20μg/m$^3$ via the two main pathways in the southwest and northeast. The ratio of near-surface dust concentration to PM10 shows a similar spatial pattern. Total dust emission was estimated to be 40 MT (mega-tonnes) per year over the period 1980-2020, of which nearly 50% was deposited on land and the rest exported away from the Australian continent.

## 1. Introduction

Dust storms as one natural disaster occur frequently in Australia, especially in the central inland area as the largest dust source in the Southern Hemisphere (Shao, 2009; McTainsh et al., 2011b; Ekström et al., 2004; McTainsh et al., 2011a), contributing to approximately 5% of the global total dust emissions (Shao, 2009; Wu et al., 2020;



Chen et al., 2022). Dust sourced from the Lake Eyre Basin is not only deposited on the Australian continent but
also transported to the Tasman Sea in the southeast and the Indian Ocean in the northwest (Strong et al., 2011;
Bowler, 1976; Sprigg, 1982; Speer, 2013; Ekström et al., 2004; Shao et al., 2007). The adverse impacts of dust
storms on populated areas include incalculable economic loss in agriculture (Stefanski and Sivakumar, 2009) and
household cleaning and associated activities (Tozer and Leys, 2013), human health issues such as respiratory
problems (Roberts, 2013; Chen et al., 2007; Cowie et al., 2010; Goudie, 2014; Middleton, 2017) and
cardiovascular disease (Domínguez-Rodríguez et al., 2021; Zhang et al., 2016), and contamination of water
sources (Middleton, 2017). Moreover, the Australian dust over the south-west Pacific Ocean strengthens the
relationship between rainfall and the El Niño Southern Oscillation (ENSO) by driving ENSO-related anomalies
in radiative forcing (Rotstayn et al., 2011), which is the direct dust feedback to climate (Shao et al., 2013).
The severity of dust activities can be indicated by visibility-based dust event days/dust storm index (DSI) (Yu et
al., 1992, 1993; McTainsh et al., 2011b; O'Loingsigh et al., 2017) and dust concentration (Shao et al., 2013;
McTainsh et al., 2005; Tews, 1996; Baddock et al., 2014), aerosol optical depth (AOD)/DAOD (dust AOD)
(Ginoux et al., 2010; Pu and Ginoux, 2018; Yu and Ginoux, 2021; Ginoux et al., 2012; She et al., 2018) and dust
index (Di et al., 2016; Yang et al., 2023; Bullard et al., 2008), simulated near-surface dust concentration (Prospero
et al., 2020; Buchard et al., 2017), and PM10 (particles with a diameter of 10 micrometers or less) (Leys et al.,
2011; de Jesus et al., 2020). The dust event database (DEDB)/dust storm index (DSI) is widely used in dust and
wind erosion research in Australia, especially for spatial and temporal variations in dust activities and has
benefited from long-term temporal data and widely distributed Bureau of Meteorology (BoM) sites (McTainsh
and Pitblado, 1987; McTainsh et al., 2011a; O'Loingsigh et al., 2014). Horizontal visibility has also been used for
estimating dust concentration and dust loading for large dust storms (McTainsh et al., 2005) and visibility-based
dust concentration has been even used for exploring the climate forcing of dust at the global scale (Shao et al.,
2013). With development of satellite remote sensing and numerical dust models, remote sensing and General
Circulation Model (GCM) products have been increasingly applied to dust research on spatial extent detection,
columnar optical properties, and near-surface concentrations. Dust index based on satellite images can be traced
back to the detection of dust storms using Advanced Very High-Resolution Radiometer (AVHRR) data (Ackerman,
1989), taking advantage of AVHRR's large spatial coverage. So far, several different dust indexes have been
developed for regional or global dust detection and different sensors (Yang et al., 2023). Satellite data retrieved



AOD/DAOD have been more frequently applied to quantitative dust research since AOD was successfully
retrieved over bright dust source areas (Hsu et al., 2004; Ginoux et al., 2010; Baddock et al., 2009). Benefiting
from satellite providing dust source schemes (Ginoux et al., 2001), and near-surface dust concentration has been
simulated for regions or at the global scale (Gelaro et al., 2017; Buchard et al., 2017; Shao et al., 2007; Wu et al.,

61  2020).


Analysis of long-term AOD/DAOD data has not been attempted for Australia. AOD/DAOD was mostly used for
identifying the spatial extent of single dust events or as reference data for evaluating dust detection algorithms in
Australia. For example, Baddock et al. (2009) assessed the performances of four detection algorithms based on
Moderate Resolution Imaging Spectroradiometer (MODIS) L1 (Level 1) B data and MODIS Deepblue (DB) AOD
for central Australia (i.e. the Lake Eyre Basin) on identifying airborne dust and mineral aerosols. There are a few
dust studies on analyzing seasonal spatial variations of dust using a multi-year AOD/DAOD dataset. Ginoux et al.
(2012) retrieved global DAOD using MODIS DB aerosol dataset from 2003 to 2009 and analyzed major
anthropogenic and natural dust emissions in Australia. Their results show that the contribution to total emissions
by anthropogenic activities can be as high as 75% in Australia. Yu and Ginoux (2021) show the monthly MODIS
DB DAOD and Multi-angle Imaging SpectroRadiometer (MISR) coarse mode AOD at 15 AERONET sites and
the annual DAOD and coarse mode AOD in Australia from 2000 to 2019. The comparison with DSI shows that
satellite AOD/DAOD presents the same dusty month/season as that by DSI at three AERONET sites in the dust
source area. Yang et al. (2021) show similar AERONET coarse mode AOD variations and seasonal contribution
of dust to total aerosols at nine AERONET sites and analyze seasonal DAOD from MERRA-2 aerosol reanalysis
from the early 2000s to 2020. There are limitations to this kind of study. Firstly, MODIS DB retrieved DAOD
shows much smaller coverage than the original DB AOD due to excluding low background DAOD, possibly
resulting in overestimation of dust activity severity. Secondly, MERRA is very likely to have underestimated
DAOD over 0.2, especially for severe dust storms such as those on the 23$^{rd}$ of October 2002 and 23$^{rd}$ of September
2009 (Che et al., 2022).

PM10 is often taken as an effective indicator of dust severity for single dust storms (Leys et al., 2011; McGowan
and Clark, 2008); however, long-term analysis of dust severity for Australia is of great difficulty using PM10 data.
First, PM10 observations in each state mostly began after 2000 in populated urban areas while the dust source



area in central Australia lacks PM10 observations. This spatial distribution of PM10 sites also causes difficulties
in retrieving PM10 in Australia using satellite AOD. Second, little progress has been made to retrieve PM10 for
large regions based on remote sensing products in Australia because 1) a reliable estimate of PM10 is difficult to
obtain due to the relatively low dust concentrations approaching its retrieval uncertainty and 2) inclusion of AOD
and related predictors cannot improve the accuracy of simulated PM10 (Pereira et al., 2017). Therefore, long-term
time series analysis of PM10 in Australia has predominantly focused on site-based observations. For example, de
Jesus et al. (2020) analyzed PM10 trends in major cities of Australia over the last two decades using site PM10
observations. Compared with PM10, near-surface dust concentration is capable of indicating the severity of dust
events without providing information on the proportion of fine particles because PM10 observations include not
only dust particles but also other types of particles. For example, Love et al. (2019) analyzed a 17-year (1990-
2007) near-surface dust concentration data collected by a high-volume air sampler (HVS) in Mildura. Long-term
dust analysis studies such as this are relatively few compared to those based on visibility-transferred dust
concentrations. However, the accuracy of visibility-transferred dust concentration cannot be ensured. As noted by
McTainsh et al. (2005),   the relationships between visibility and dust concentration obtained in the United States
of America (USA) (Chepil and Woodruff, 1957) and Australia in the 1990s (Tews, 1996) have been inappropriate
for estimating dust concentration over different areas of Australia, since visibility-based dust concentrations are
strongly influenced by dust particle size,.

Development of numerical dust models and GCMs provides dust cycle simulations for understanding the impacts
of dust on the earth systems. Models such as the Georgia Tech/Goddard Global Ozone Chemistry Aerosol
Radiation and Transport (GOCART) (Ginoux et al., 2001), Mineral Dust Entrainment and Deposition (DEAD)
(Zender, 2003), and Aerosol Species IN the Global AtmospheRe (MASINGAR) coupled with MRI/JMA 98 GCM
(Tanaka and Chiba, 2006) are all capable of providing dust cycles for regions around the world. However, dust
emissions and depositions vary substantially among models (Chen et al., 2022; Wu et al., 2020). This would
directly lead to a large discrepancy in conclusions based on different models. For example, the contribution of
Australian dust to global dust emissions is estimated to vary from 0.02% to 27.8% using simulation outputs from
15 CMIP5 (Coupled Model Intercomparison Project Phase 5) models (Wu et al., 2020). Therefore, long-term
analysis of the dust cycle in Australia needs a dataset with high accuracy and capability to quantify long-term
trends and variabilities.




MODIS DB and MERRA-2 data products, dust aerosols, and near-surface dust concentration/PM10 observations
were analyzed in this study for a better understanding of long-term dust entrainment and transport over Australia.
Objectives of this study include:
1. To develop a DAOD dataset using MERRA-2 aerosol reanalysis and MODIS DB aerosol datasets;
2. Validate MERRA-2 near-surface dust concentrations using ground-based visibility data sets, and

MERRA-2 estimated PM10 with ground-based PM10 observations sourced from the New South Wales

Air Quality Monitoring Network (NSW AQMN);

3. To corroborate MERRA-2 dust flux with major dust pathways identified in previous studies;
4. To map the seasonal MODIS and MERRA-2 (M&M) DAOD from 2002 to 2020 and seasonal MERRA-

2 near-surface dust concentrations/PM10 over the period from 1980 to 2020;

5. To quantify the annual dust cycle for Australia over the period from 1980 to 2020, including dust emission

in Australia using MERRA-2 emission data, dust import and export using MERRA-2 flux data, and dust

deposition using MERRA-2 emission and flux data.

**2. Data and methodology**
**2.1 Ground-based PM10 and AERONET data**
AOD at 440nm and Ångström exponent (AE) at 440-675nm from AERONET v3 solar products were used for
calculating AOD at 550nm, as well as Level 1.5 (L1.5) single scattering albedo (SSA) at 440nm from v3 inversion
product to retrieve DAOD. The latest AERONET v3 solar product includes data at Level 1.0 (L1.0) (without data
screening), L1.5 (with cloud screened and quality controlled), and Level 2.0 (L2.0) (quality assured)
(https://aeronet.gsfc.nasa.gov/). Giles et al. (2019) reported that AOD from the AEROENT V3 product was with
a low uncertainty, suggested by a bias of +0.02 and one sigma uncertainty of 0.02. Since satellite AOD normally
refers to that at 550nm, AE is necessarily used for spectrally interpolating AOD to this wavelength according to
the dependence of AOD on wavelength (Angstrom, 1924). The Version 3 (V3) inversion product also includes
data at three levels, L1.0, L1.5, and L2.0. The main difference between L1.5 and L2.0 is that the L2.0 inversion
product is only made when the corresponding AOD is higher than 0.4 (Dubovik and King, 2000). This leads to
the data volume of L2.0 SSA being far smaller than that of L1.5, especially over Australia predominated by low
AOD conditions. L1.5 SSA data with a much larger data volume, therefore, was used for identifying dust-



contaminated AOD. In this study, all L2.0 AOD and AE, L1.5 SSA, as well as fine mode fraction (FMF) in the
Spectral Deconvolution Algorithm (SDA) database in Australia from 1997 to 2020 were used for retrieving DAOD
at 550nm (yellow stars in Fig.1).

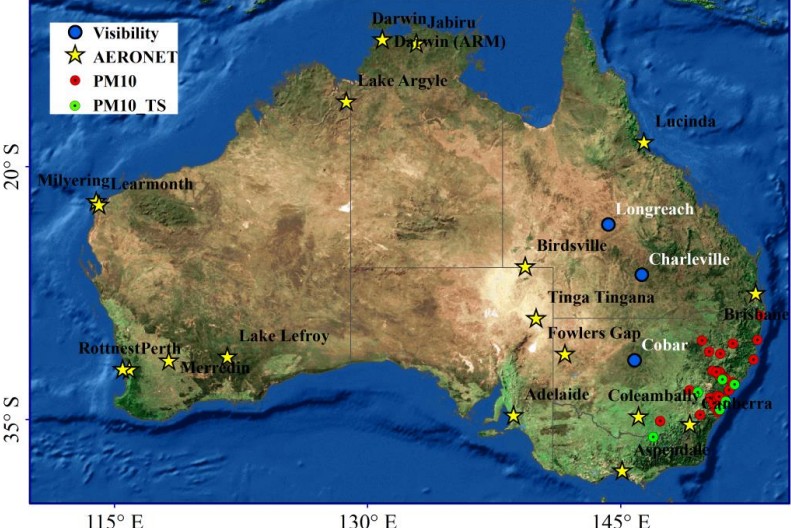

**Figure 1**: **Distribution of ground-based sites. Yellow stars are inland AERONET sites in Australia; Green**
**and red circles indicate PM10 observation sites for validating MERRA-2 products, among them time series**
**analysis was conducted at green sites; Circles in blue are visibility observation sites.**

Horizontal visibility records from the BoM were manually estimated and noted by a weather observer. All
visibility records have an upper limit of 10km, which means that if the record shows a visibility of 10km the actual
visibility could be 10km or greater. In addition to visibility, a synoptic code was recorded during notable weather
events such as dust storms (Baddock et al., 2014) as well as a weather type for the weather observation
immediately prior to the current observation (O'Loingsigh et al., 2010). There are 11 SYNOP (surface synoptic
observations) codes for dust weather, including dust haze, raised dust or sand, dust whirls, thunderstorm with sand
or dust storms, dust storms, and so on. Due to the visibility record only allocated one SYNOP code, the highest-
coded weather code was retained although it may have included several weather types (O'Loingsigh et al., 2010).
The long-term visibility records with SYNOP codes have been widely used for wind erosion research (McTainsh
et al., 1989b, 1990, 2011a), dust event climatologies including the DEDB and DSI developed at Griffith University



(McTainsh et al., 2011b; O'Loingsigh et al., 2014, 2010) and single dust events (Shao et al., 2007; McTainsh et
al., 2005; Leys et al., 2011). In this study, all hourly BoM horizontal visibility data with a dust type SYNOP code
(excluding thunderstorms) were used for validating MERRA-2 near-surface dust mass concentration at three sites
(Charleville, Cobar, and Longreach, Fig.1) from 1980 to 2020.

PM10 concentrations are publicly downloadable from the NSW AQMN website (https://www.dpie.nsw.gov.au/).
AQMN equips a Tapered Element Oscillating Element instrument (TEOM), measuring atmospheric particles <
10μm. The filter in the TEOM weighs collected samples every 2 seconds and the average value is reported hourly.
Data quality control is applied to all PM10 databases according to Australian Standard 3580.9.8 (Leys et al., 2011).
All PM10, gases, and climate observations data can be accessed using the AQMN web data download facility
(https://www.dpie.nsw.gov.au/air-quality/air-quality-data-services/data-download-facility). In this study, monthly
PM data from 62 AQMN urban sites (circles in green and red in Fig.1) were used for validating MERRA-2 near-
surface dust concentrations from 2001 to 2020.
**2.2  MERRA-2 aerosol reanalysis**
MERRA-2 aerosol reanalysis provides long-term global aerosol parameter datasets from 1980 to the present
(Randles et al., 2017; Gelaro et al., 2017; Buchard et al., 2017). MERRA-2 includes optical depth, near-surface
mass concentrations, column mass concentrations, two directional mass flux for each aerosol component,
including sea salt, sulfate in (SO4 and SO2), organic carbon, dust, and black carbon. Due to the inclusion of the
GEOS and GSI assimilation system, MERRA-2 aerosol simulations perform comparably with high-quality
satellite-based datasets and are fairly close to aerosol observations. Benefiting from the incorporation of space-
based and ground-based observations, the accuracy for the total AOD is guaranteed with physical models outputs
constrained              by              the              assimilation              system
(https://gmao.gsfc.nasa.gov/research/science_snapshots/2015/MERRA2_global_aerosol_dist.php). In this study,
MERRA-2 dust (DAOD) and total AOD were used for providing the ratio of dust aerosols over total particulate,
all near-surface aerosol mass concentrations for estimating PM10, as well as, dust flux for estimating dust loading
import/export for Australia.

Numerous validation studies have shown that MERRA-2 optical depth and near-surface concentrations could be
used for temporal and spatial analysis and even long-term analysis of aerosols regionally or globally due to the



high quality and long temporal coverage from 1980 to the present. This kind of study primarily focuses on the
validation of MERRA-2 AOD with ground-based AOD from AERONET (Buchard et al., 2017; Sun et al., 2019a;
Che et al., 2022; Randles et al., 2017), SONET (Sun-Sky Radiometer Observation Network) (Ou et al., 2022),
SKYNET (Sun et al., 2019b). For dust research in Australia, MERRA-2 AOD has been validated/evaluated with
AERONET (Che et al., 2022; Mukkavilli et al., 2019) and MODIS DB dataset (Che et al., 2022), as well as its
DAOD with the MACC (Monitoring Atmospheric Composition and Climate) simulation (Mukkavilli et al., 2019)
and MODIS DB retrieved DAOD (Che et al., 2022). However, few studies were carried out to validate MERRA-
2 near-surface dust concentrations.

Limited by a lack of ground-based near-surface observations, most validation studies of MERRA-2 component
concentrations focus on a single site or a few sites, especially the dust component. For example, MERRA-2 OC,
BC, sulfate concentration data sets have been validated against ground-based observations in China in Nanjing
(Zhao et al., 2021) and Jingsha and Lin'an (Ma et al., 2021), Beijing (only BC) (Qin et al., 2019; Ou et al., 2022)
and over northern India (OC and BC) (Soni et al., 2021). As for the dust component, daily MERRA-2 dust
concentrations have been validated in Barbados (daily product) (Buchard et al., 2017), and Cayenne, Northern
South America (Prospero et al., 2020).

Although there are relatively few ground-based observations, validation results still show daily and monthly
MERRA-2 surface mass concentrations in the forms of PM2.5/10 or single components with relatively high
accuracy. MERRA-2 near-surface component concentrations can be used for conversion to PM10 using the
equation below (Provençal et al., 2017; Ma et al., 2021):

$$[\text{PM10}] = 1.375 \times [\ SO_4^{2-}] + 1.8 \times [OC] + [BC] + [DU_{10}] + [SS_{10}] \qquad (1)$$

where $[SO_4^{2-}]$, $[OC]$, $[BC]$, $[DU_{10}]$, and $[SS_{10}]$ are concentrations of each aerosol component, namely sulfate,
organic carbon, black carbon, dust, and sea salt aerosol, and the subscripts 10 indicates the particle diameter less
than 10μm. Ma et al. (2021) derived from using the first three size bins for dust and sea salt aerosols. $[SO_4^{2-}]$ is
multiplied by 1.375 under the assumption that $SO_4$ is fully neutralized by ammonium in the form of $(NH_4)_2SO_4$
(ammonium sulfate) and a scale factor of 1.8 for OC is included to take into consideration the organic compounds
in the particulate organic matter. Equation 1 was developed for estimating MERRA PM10 over Europe (Provençal
et al., 2017), which may be inappropriate for PM10 estimation over other regions. For example, Ma et al. (2021)



considered the increasing trend of nitrate which was a large proportion of aerosols over China and revised Eq.1
as:

$$[PM10] = 1.375 \times [SO_4^{2-}] + 1.29 \times [NO_3^-] + 1.8 \times [OC] + [BC] + [DU_{10}] + [SS_{10}] \qquad (2)$$

where $[NO_3^-]$ is the concentration of nitrate. Considering that coarse mode aerosols take up 57%-71% of total
aerosols over major cities (Chan et al., 2008), MERRA-2 PM10 for Australia was estimated using Eq.1.

In this study, the method developed by GMAO (Eq.3) using MERRA-2 3-D aerosol mass mixing ratios was used
for PM10 estimation over Australia.

$$[PM10] = (1.375 \times [SO_4^{2-}] + [BCphobic] + [BCphilic] + [OCphobic] + [OCphilic] + [DU001] + \qquad (3)$$

$$[DU002] + [DU003] + 0.74 \times [DU004] + [SS001] + [SS002] + [SS003] + [SS004]) * AIRDENS$$

where the subscripts philic and phobic for [BC] and [OC] represent hydrophilic and hydrophobic BC and OC
aerosols, respectively, the numbers after [DU] and [SS] indicate the 4 size bins, i.e. 001 to 004 represent the bins
with radius from 001 for 0.1 to 1.0μm, 002 for 1.0 to 1.8μm, 003 for 1.8 to 3.0μm, and 004 for 3.0 to 6.0μm for
dust, and similarly for 0.03 to 0.1μm, 0.1 to 0.5μm, 0.5 to 1.5μm, and 1.5 to 5.0μm for sea salt, and the AIRDENS
means air density (https://gmao.gsfc.nasa.gov/reanalysis/MERRA-2/FAQ/#Q5)
**2.3 MODIS DeepBlue AOD dataset**
MODIS DB aerosol product provides nearly full global coverage of AOD, AE, and SSA datasets over EOS (Earth
Observing System) years (from 2000 to the present) (Sayer et al., 2017). MODIS DB has been widely used for
dust research over arid and semi-arid regions i.e., bright surfaces where the traditional dark target (DT) algorithm
is not applicable. For example, Ginoux et al. (2012) analyzed the global distribution of dust sources and emissions
using MODIS DB aerosol product, taking advantage of its coverage over bright surfaces and successful retrieval
of AE and SSA together with AOD. Similarly with other aerosol products released by NASA like DT (Levy et al.,
2013) and aerosol climate change initiative (aerosol_CCI) such as AATSR (Advanced Along-Track Scanning
Radiometer) aerosol products (de Leeuw et al., 2015; Sundström et al., 2012; Kolmonen et al., 2016; Thomas et
al., 2009), all L2 MODIS DB aerosol datasets in the latest C61 product were produced with a spatial resolution of
10km with all MODIS radiance data. In this study, MODIS DB aerosol product for Aqua from 2002-2020 was
selected.

MODIS DB key parameters have been validated over Australia and globally, especially AOD. The MODIS DB



AOD dataset has been validated against AERONET data (Che et al., 2022; Sayer et al., 2019; Wei et al., 2019),
inter-compared with other AOD products for MODIS, such as MAIAC and DT (Shaylor et al., 2022), and even
evaluated with MERRA-2 AOD (Che et al., 2022) over Australia. These studies show that there is a high
probability of data points (MODIS DB and AERONET) within EE lines, such as 67% points with an EE of
$\pm(0.03 + 0.15\tau)$ (Che et al., 2022). The latest MODIS DB aerosol product limits AE values from 0 to 1.8, and
in low AOD conditions AE is set to < 1.0 over bright surfaces and AE is fixed to 1.5 over vegetated surfaces (Sayer
et al., 2013; Hsu et al., 2013). Sayer et al. (2019) tested the performance of MODIS AE in different conditions,
including dust cases and fine mode cases. Over vegetated surfaces, MODIS DB AE was overestimated
systematically with a broad range of error from 0.5 to 1 for dusty conditions, while over dry surfaces the
performances of MODIS DB AE have improved in systematic overestimation but still a broad error range (Sayer
et al., 2019). Therefore, dust detection by MODIS DB could be uncertain to some extent.
**2.4  Gridded SILO monthly rainfall data**
The SILO datasets developed by the Queensland government are aimed at providing long-term continuous point
and full coverage gridded climate datasets for land areas of Australia from 1989 to the present
(https://www.longpaddock.qld.gov.au/silo/). The full coverage gridded rainfall dataset at a temporal resolution of
a day and a month were produced by interpolating BoM daily and monthly rainfall observations with a Kriging
method (Jeffrey et al., 2001). Validation results show that the accuracy of SILO data is typically higher around
areas with densely distributed BoM rainfall gauge sites (Jeffrey et al., 2001). Overall, except for parts of western
Australia with few BoM rainfall sites SILO rainfall shows a high accuracy with an $R^2$ >0.8 over most of the
Australian land (Jeffrey et al., 2001).
**2.5 MODIS-MERRA (M&M) combined DAOD**
DAOD has been successfully retrieved based on MODIS DB product by a coarse mode fraction (CMF) in
Equation 4 (Ginoux et al., 2010, 2012; Pu and Ginoux, 2017):

$$CMF = 0.98\text{-}0.5098AE\text{-}0.05AE^2 \tag{4}$$

The early version of the MODIS DB DAOD dataset has been used for analyzing the global distribution of dust
sources (Ginoux et al., 2012) and constructing a global DAOD climatology (Voss and Evan, 2020). The new
version of MODIS DB DAOD dataset has been used for evaluating DAOD satellite remote sensing from GCMs
and CALIOP, such as CMIP5 (Pu and Ginoux, 2018), MERRA-2 (Che et al., 2022), and CALIOP (Song et al.,



2021). In Australia, the MODIS DB DAOD dataset was validated against AERONET data and results show that
88% and 71% of data points for MODIS/Terra and MODIS/Aqua, respectively, fall within an EE of ($\pm0.05+0.15\tau$)
(Che et al., 2022). Although studies have shown MODIS DB DAOD dataset is of high quality, there are still
several factors limiting its applications to dust research in Australia. First, the data coverage of the MODIS DB
DAOD dataset only includes obvious dust plumes. Secondly, MODIS DB AE retrievals have a broad range of
errors over both bright and vegetated surfaces, especially systematical overestimations over vegetated surfaces
(Sayer et al., 2019), causing non-negligible uncertainty in the MODIS DB DAOD dataset. Thirdly, AE was fixed
in low AOD conditions (Sayer et al., 2013; Hsu et al., 2013). MERRA-2 aerosol reanalysis is expected to make
up for these deficiencies.

A new DAOD dataset has been developed for Australia in this study using MODIS DB aerosol product and
MERRA-2 aerosol reanalysis. Figure 2a shows MODIS DB AOD on 23$^{rd}$ October 2022 with hundreds of
kilometers of dust plume in eastern Australia while the dust plume is seriously underestimated by MERRA-2
(Figure 2b) compared with MODIS BTD in Fig.2c. In order to take advantage of MODIS DB in catching dust
plumes and MERRA-2 in spatial coverage, DAOD equals to MODIS DB DAOD when it is available, otherwise,
DAOD equals to MODIS DB AOD multiplied by the ratio of MERRA-2 DAOD to total AOD. This is based on
the assumption that the dust fraction in MERRA-2 is shown to have high accuracy with LIVAS (Gkikas et al.,
2021) and AERONET (Che et al., 2022). Figure 2d shows that the final DAOD is capable of screening sea salt
AOD over the Cape York Peninsula and has the same spatial coverage as MODIS DB AOD. Although Sayer et al.
(2019) suggest that AE should be only used for discriminating coarse mode-dominated AOD from fine mode-
dominated AOD qualitatively, a smoke plume (Fig.2d) over the Australian east coast was effectively removed
from MODIS DB AOD.

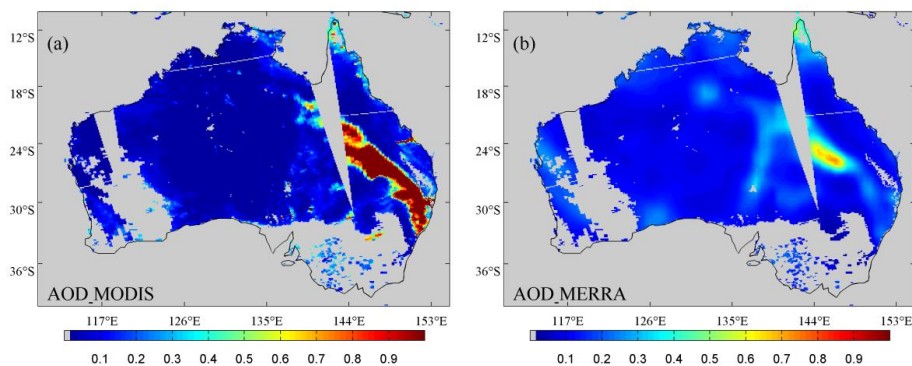




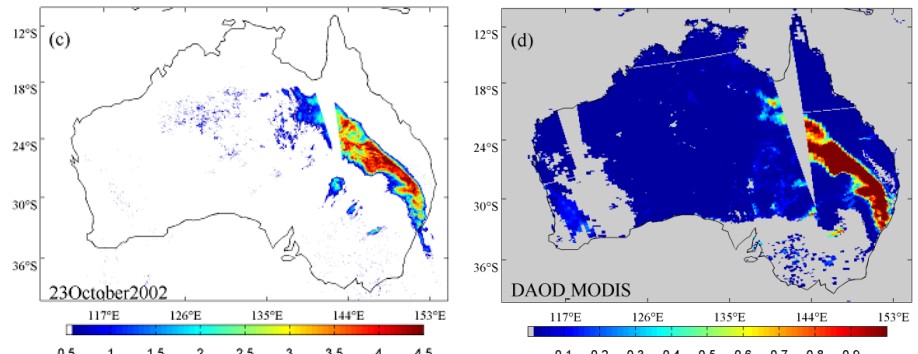

**Figure 2: Development of DAOD using MERRA-2 and MODIS DB dataset. (a) MODIS DB AOD, (b) MERRA-2 AOD, (c) MODIS BTD, and (d) MERRA-MODIS combined DAOD (M&M).**

## 3.   Results

### 3.1  Validation of MERRA-2 surface mass concentration and DAOD

Figure 3 shows the validation result of MODIS (Aqua)-MERRA (M&M) combined DAOD over Australia from 2002-2020. The average AERONET DAOD for all collocated data points is only 0.03 for Australia. When AOD is low, remote sensing AOD retrievals are likely to be close to the margin of uncertainty and hence subject to large relative bias, especially over the arctic region (Mei et al., 2013b, a), Qinghai-Tibet Plateau (Che et al., 2018, 2016), and Australia (Che et al., 2022; Sayer et al., 2019). The ratio of RMSE (root mean square error) to the mean AERONET DAOD for MODIS-MERRA DAOD was 0.8, indicating that the uncertainty is close to MODIS-MERRA DAOD. The EE (expected envelope) that contains 68% of data points is $\pm(0.016 + 0.15\tau_{Aero})$ for MODIS-MERRA DAOD to AERONET DAOD over Australia. The intercept in EE is 0.016 is much smaller than for the MODIS DB AOD over Australia (0.03) (Che et al., 2022), suggesting a high level of accuracy of this MODIS-MERRA DAOD dataset with a smaller absolute error.



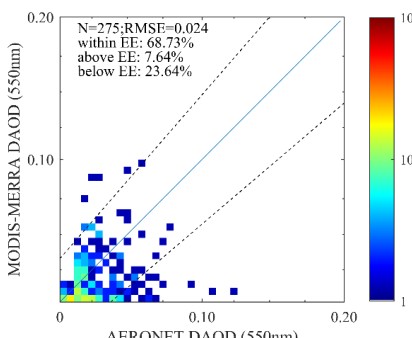


**Figure 3: Comparison of MERRA-MODIS DAOD with AERONET DAOD. The dashed lines denote an EE**

**of $\pm(0.016 + 0.15\tau_{Aero})$ which contains 68% of the data points. $\tau_{Aero}$ represents AERONET AOD.**

Figure 4 shows the validation results of MERRA-2 estimated monthly PM10 with ground-based observations at
62 AQMN stations for the period 2001-2020. There are 7438 data points in total for MERRA-2 PM10 validation
over eastern NSW. These selected PM10 observations are located downwind areas of inland dust sources such as
Lake Eyre, South Simpson lakes and the Channel country (O'Loingsigh et al., 2017). PM10 at these sites, therefore,
could represent dust activities in southeast Australia during dust seasons. The mean monthly PM10 for all data
points is 19.9μg/m³, indicating a clear atmosphere over eastern NSW on average. When PM10 observations are
greater than 40μg/m³, almost all the data points are below the 1-1 line, suggesting that MERRA-2 is incapable of
catching high PM10 events (dust or other pollutions). Due to the relatively dense spatial distribution, AQMN sites
are likely to observe the same dust events with similar PM10 observations and thus a similar extent of
underestimation. Time series plots (Fig.5) show similar severe underestimation in Newcastle (13.5μg/m³ vs.
106.9μg/m³), Randwick (12.4μg/m³ vs. 84.7μg/m³), and Wollongong (13.0μg/m³ vs. 65.2μg/m³) in September
2009 and Bathurst (18.9μg/m³ vs. 104.8μg/m³), Bulga (30.4μg/m³ vs. 78.9μg/m³), and Newcastle (37.1μg/m³ vs.

50.8μg/m³) in December 2019. In September 2009, a severe dust storm swept the Australian continent,

causing a jump in PM10 for overpass areas (Leys et al., 2011). High PM10 (higher than 300μg/m³) lasted for
approximately 12 hours and reached as high as15388μg/m³ at Bathurst. Similarly, high PM10 concentrations were
recorded at Bulga and Newcastle (Fig.5). Due to rainfall deficiency and high temperatures in November and
December 2019, NSW experienced the longest bushfire season when more than five million hectares were burned
(BBC News, 2020). The NSW AQMN stations, therefore, had recorded high PM10 concentrations during the
bushfire season. These underestimations by approximately 5 were also a major reason why the ground-based mean



PM10 (19.9μg/m$^3$) was higher than that of MERRA-2 (18.2μg/m$^3$). When PM10 is less than 20μg/m$^3$, data points
show a slight bias of MERRA-2, and when PM10 is between 20μg/m$^3$ to 40μg/m$^3$, the bias in MERRA-2 reduces.
This is shown by evenly distributed data points around 1-1 line but less association occurs between the two. Severe
underestimations of MERRA-2 PM10 (Fig.5) show a strong seasonality at some sites like Albury and Bathurst in
summer when PM10 is greater than 40μg/m$^3$. This suggests that MERRA-2 is very liky to underestimate dust
severity in summer because dust events mainly occur in summer throughout the year in NSW (Che et al., 2022).
In spite of the underestimations, MERRA-2 is capable of tracing the seasonal variations of PM10 at six sites with
an r$^2$ value from 0.14 to 0.44. Overall, a RMSE is 10.8μg/m$^3$ for all monthly MERRA-2 and ground-based PM10
data.

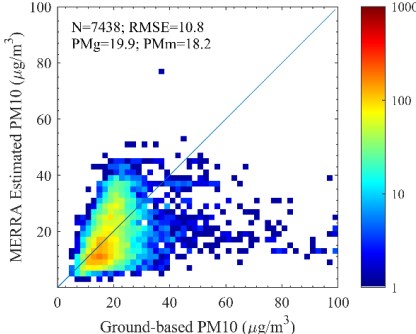


**Figure 4: Comparison of monthly MERRA-2 estimated PM10 with ground-based PM10 observations at 62**
**AQMN stations.**

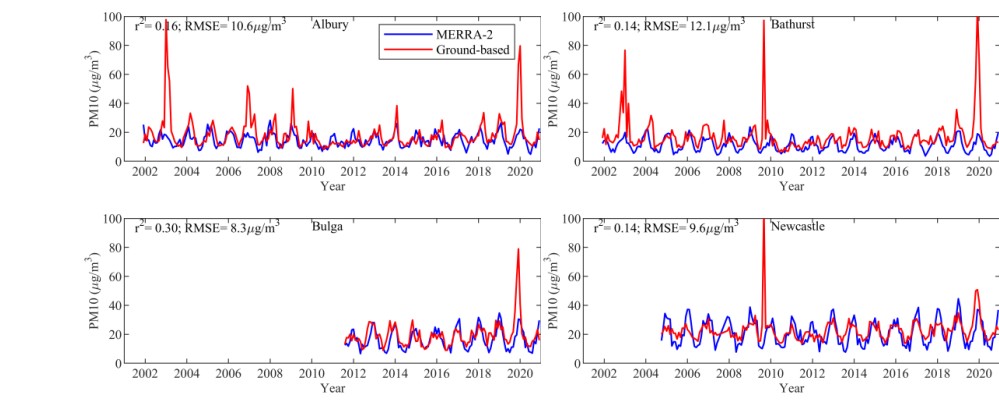





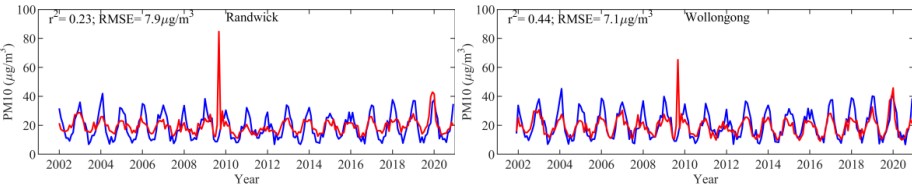


**Figure 5: Time series for MERRA-2 estimated PM 10 with ground-based PM observations at 6 sites in NSW.**


Figure 6 shows the relationships between MERRA-2 near-surface dust concentrations with horizontal visibility
with a dust type based on SYNOP code at Charleville, Cobar, and Longreach. The relationships are similar at
Charleville and Cobar in that MERRA-2 near-surface dust concentrations follow a power function relationship
with horizontal visibility. The $r^2$ values for two sites of 0.44 and 0.39, respectively, also suggest a relatively robust
relationship between the two datasets. At Longreach, a low $r^2$ value of 0.18 shows a weak relationship between
MERRA-2 near-surface dust concentrations and visibilities. Longreach is known to bein a region of frequent local
wind erosion activities with extensive tracts of clay soils in Eastern Australia (McTainsh et al., 1990). Alluvial
sediments and sandy clays, therefore, would be important sources for local dust events in Longreach (Rust and
Nanson, 1989; McTainsh et al., 1990). These clay aggregates   exhibit lower optical extinction compared to fine
clay. However, due to their larger mass, they can still contribute to a high concentration of dust near the surface
in a high visibility condition. This explains why the r2 in Figure 6c differs from the other two. Previous studies
showed that near-surface dust concentrations/total suspended particle concentrations agree statistically well with
the visibility data (Baddock et al., 2014; Chepil and Woodruff, 1957; Shao et al., 2003; Tews, 1996) and visibility-
defined DSI (O'Loingsigh et al., 2014), and horizontal visibility were often used for calculating dust
concentrations (Leys et al., 2011; McTainsh et al., 2005). According to Shao et al. (2007),   visibility observations
are subjective and inaccurate; the sample size for regression equations is usually small; and   dust concentrations
are affected by dust particle size and humidity. Similar power function relationships between the two suggest the
acceptable accuracy of MERRA-2 near-surface dust concentrations to an extent.






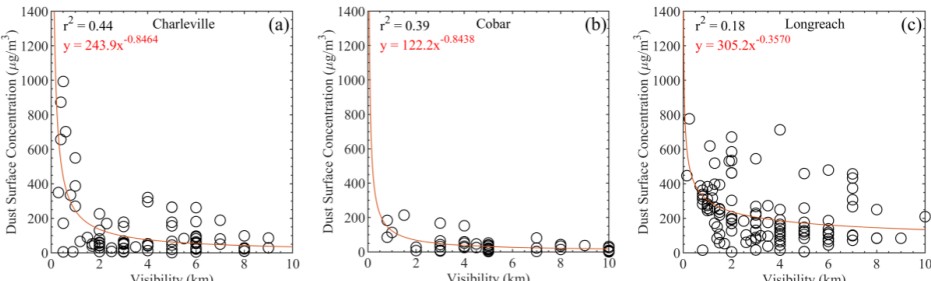

**Figure 6: Relationships between BoM horizontal visibility observations of MERRA-2 dust surface mass concentrations.**

Figure 7 shows dust pathways and sink areas for Australia as identified in previous studies. Bowler (1976) established the first dust pathways (Figure 7a) using trends of dust and sand dune movement during the period of intense dune building phases. Major dustfall areas lie mainly to the southeast and northwest of the Australian continent. Sprigg (1982) proposed a conceptual model (Figure7b) for describing how wind systems fed dust into the pathways identified by Bowler using measured wind run, wind direction, and wind speed in desert areas. Blewett (2012) adopted the dust pathways established by Bowler but provided a detailed classification of sand dunes and accurately confirmed dustfall in the southeast offshore area. Figure 7d shows MERRA-2 dust flux over Australia from 2002 to 2020. Compared to previous studies (i.e. Figure 7a-c), MERRA-2 flux quantitatively shows the mean dust pathways and dustfall areas for Australia, providing independent support to previous conceptual models.

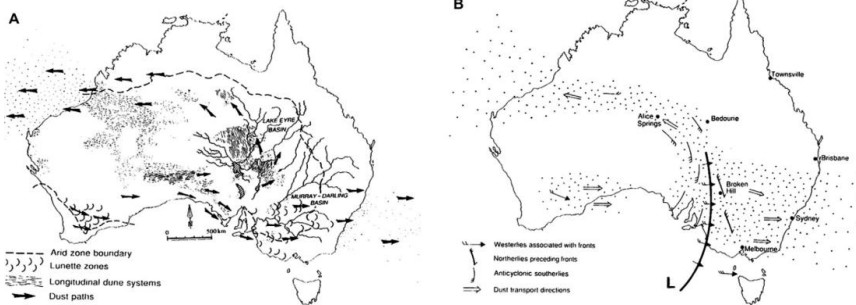





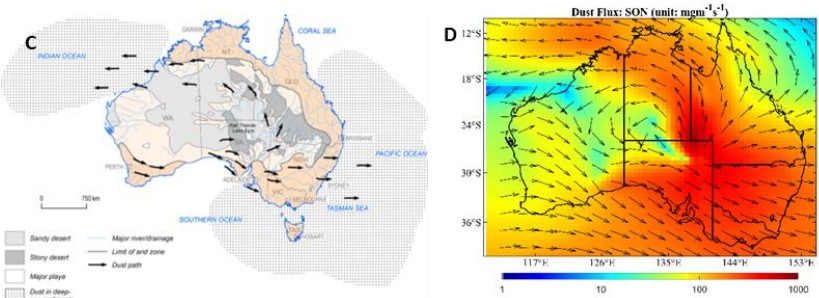


**Figure 7: Comparison of dust pathways over Australia delineated in previous studies. (a)    Bowler (1976),**
**(b) Sprigg (1982), (c) Blewett (2012), and (d) mean dust flux based on MERRA-2 from 2002-2020.**
**3.2  Seasonal DAOD based on MODIS-MERRA, dust concentration and PM10 based on MERRA-2**
Figure 8 shows the mean seasonal MODIS-MERRA DAOD over Australia from 2002 to 2020. Spring (Sep-Nov)
and summer (Dec-Feb) are typically dust seasons in Australia, when DAOD is much higher than that in autumn
(Mar-May) and winter (Jun-Aug). This seasonal pattern is consistent with that shown by monthly dust event
frequency based on synoptic observations (McTainsh et al., 1998). As dust activities would normally last until
March in southeastern Australia (McTainsh et al., 1998), DAOD in this area is relatively high compared to that in
Western Australia (Fig. 8b). High rainfall may inhibit occurrences of dust events, therefore, Figure 8c shows low
DAOD for most of Australia in winter.

High DAOD is shown over dust source areas and along main pathways. Dust and sand particles originated from
the inland area (around the Lake Eyre Basin), and were transported along two main dust transport pathways to the
north and southeast (Bowler, 1976; De Deckker, 2019; McGowan et al., 2000; Sprigg, 1982; Strong et al., 2011).
In spring (Figure 8d), high DAOD regions are mainly concentrated around the Lake Eyre Basin in southeastern
Australia, and in the southwest of Western Australia. Fig. 8d shows that the flux with dust entrained from the
source region in Central Australia to the southeast is so far the strongest over the Australian continent. DAOD and
dust flux are consistent with each other and both reflect the major dust pathway in southeastern Australia. High
DAOD can be also found in the southwest of Western Australia, which has been identified as the starting point of
the major pathways flow in the south in previous studies (Sprigg, 1982). Due to onshore winds, high DAOD
around this region is very likely to be generated from local dust sources. Also, DAOD in spring is much higher
than in other seasons in this region. In summer (Figure 8a), the highest DAOD regions were found in the main



pathway to the north as highlighted with a red box, and around the center of the Lake Eyre Basin. The high DAOD
region in the north of NT and QLD is in the main pathway, with a spatial pattern consistent with prevailing wind
directions. Another region with high DAOD in the middle of QLD shows a different spatial pattern with MERRA-
2 dust flux, which may be caused by differences in data coverage between MODIS DB and MERRA-2 datasets.
The second highest DAOD regions are concentrated around the southeast dust pathway in NSW. Meanwhile, dust
flux for these regions is much lower than those for northern regions. In autumn (Figure 8b), DAOD is an extension
of that in summer that DAOD shows a very similar spatial pattern to that in summer but lower in value. In regions
with high DAOD, DAOD distribution is consistent with that of dust flux (i.e., high DAOD corresponds to large
dust flux). In winter, dust emissions in the center of Lake Eyre Basin are the smallest and DAOD is the lowest
among the four seasons. Dust is mainly transported to the east coast of Australia, deposited in eastern Australia
and the ocean aresa to the southeast and northwest.


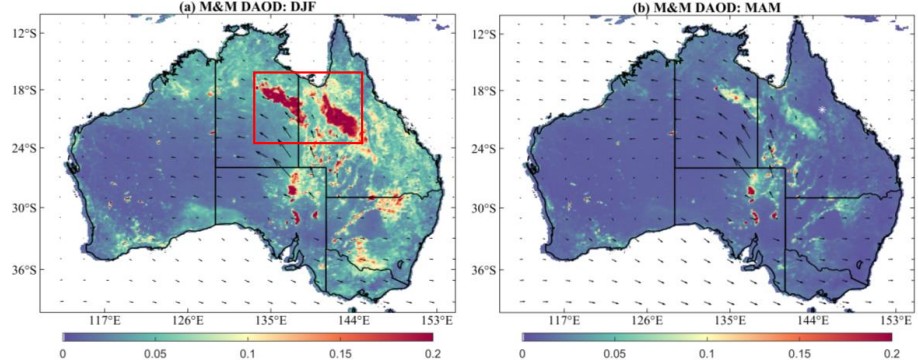



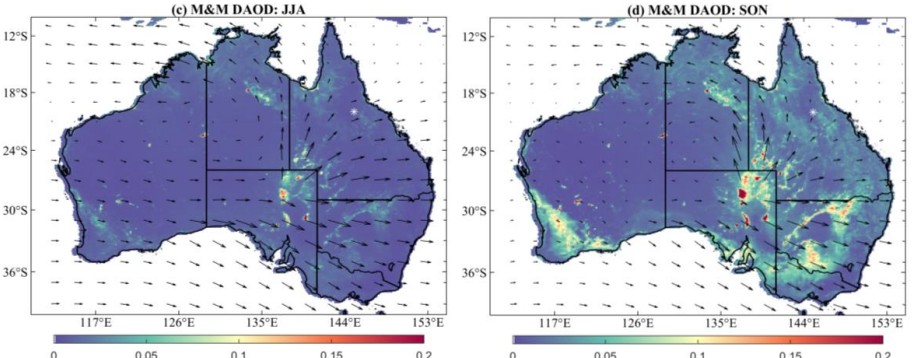




**Figure 8: Seasonal MERRA-MODIS DAOD from 2002-2020. (a) Sep-Nov, (b) Dec-Feb, (c) Mar-May, (d)**
**Jun-Jul. Arrows represent dust flux (unit: kg/m·s) with the direction and magnitude.**

Figure 9 shows mean seasonal near-surface dust concentrations and PM10 based on MERRA-2, and the ratio of
the two for Australia. The highest near-surface dust concentrations are mainly distributed over the Lake Eyre
Basin which is the largest natural dust source in Australia, while high concentrations from 50 to 100μg/m³ are
found around the Lake Eyre Basin, Great Victoria Desert, and Nullarbor Plain in four seasons (See Figure.9a, d,
g, and j). In other regions, near-surface dust concentrations typically are less than 50μg/m³ and the concentration
decreases towards the coastline. Low concentrations of less than 20μg/m³ of dust can be found over the ocean,
particularly the Indian Ocean in the northwest. MERRA-2 PM10 (See Figure 9b, e, h, and k) shows that the spatial
distribution is similar to that of the near-surface dust concentrations over the continent. Differences between the
two mainly occur in the offshore areas due to the influence of sea salt aerosols and carbonaceous aerosols in the
north in spring (Yang et al., 2021). Spatial distributions of dust concentration and PM10 are similar because PM10
accounts for the majority of dust particles in inland Australia (Figure 9c, f, I, and l). A high ratio of dust
concentration over PM10 of over 0.7 is mainly found in inland areas and the ratio decreases towards the coastline
which is consistent with that coarse particles settle out of suspension more quickly compared to finer particles
(Fryrear et al., 1991). It should be noted that four states in eastern Australia are affected relatively less by dust
than other regions from the perspective of the ratio of dust to PM10. The least affected states are VIC (higher than
0.2), NSW (0.3), and QLD (higher than 0.3), respectively.

The center of the Lake Eyre Basin emits most of the dust and largely determines the distribution of dust over the
entire continent. The near-surface dust concentration is over 300μg/m³ in the center of the Lake Eyre Basin in
spring and summer while it decreased to approximately 200μg/m³ in autumn and winter. The high concentration
is related to dust activities in the dust source areas. With frequent dust activities (in addition to high dust
concentrations over the center of the Lake Eyre Basin) the region with a dust concentration over 50μg/m³ in
summer and spring is much higher than that in autumn and winter, especially the Great Victoria Desert and
Nullarbor Plain. The near-surface dust concentrations over two major dust pathways changed a little with season
compared to that over the main dust source areas in the center of the continent. Relatively high concentrations of
near-surface dust is found to the north in the four seasons compared to that along the dust pathway to the southeast.





The wind systems responsible for dust pathways as described by Sprigg (1982), explain that pre-frontal anti
cyclonic northerly winds are responsible for the main dust pathway in the south. High dust concentrations are,
however, not found in the main dust deposition area in the southeast.




**Figure 9: Seasonal MERRA-2 near-surface dust concentration in (a) Dec-Feb, (d) Mar-May, (g) Jun-Jul,**
**and (j) Sep-Nov from 2002-2020; Seasonal MERRA-2 PM in (b) Dec-Feb, (e) Mar-May, (h) Jun-Jul, and**



**(k) Sep-Nov from 2002-2020; and the ratio of MERRA-2 near-surface concentration to MERRA-2 PM10**
**in (c) Dec-Feb, (f) Mar-May, (i) Jun-Jul, and (l) Sep-Nov from 2002-2020.**

**3.3 Dust loading budget in the Australian continent**
Figure 10 shows the directions of dust flux using MERRA-2 dust flux datasets. The colors of the coastline indicate
whether dusts are exported (red) or imported (blue). As the "loneliest" inhabited continent, Australia is located far
away from other continents, with the largest natural dust source in the southern hemisphere , the Lake Eyre Basin,
and thus regarded as a main dust source exporting dust. However, the blue border shows that areas of import can
occur along the west, north, and south coasts. Along the north coast, exported dust from QLD could be transported
back to the Cape York Peninsula, part of this dust would travel on to Arnhem Land and even travel back to the
continent. This recirculated dust cannot be defined as imported dust because it originated from the Australian
continent, was transported over the sea and back onto the continent. A similar situation can be found on the
coastline in South Australia where dust originated from the continent and was transported outwards from the
Nullarbor Plain, across the Spencer Gulf and back to South Australia and VIC. Different from these two situations,
dust imported from the west coast is very likely to be from remote dust sources in South Africa. Firstly, MERRA-
2 dust flux doesn't show dusts exported from the northwest coast are transported back to the continent. Secondly,
dust originating from the Mallee region is unlikely to be transported to the west coast crossing the Pacific Ocean
(Bhattachan and D'Odorico, 2014). Thirdly, the southwesterly winds are key to transporting dust from South
Africa to Western Australia due to their similar latitudes (Torre et al., 2022). The dust imported from 20ºS to 36ºS
is therefore regarded as the only imported dust from an external source for Australia in this study.

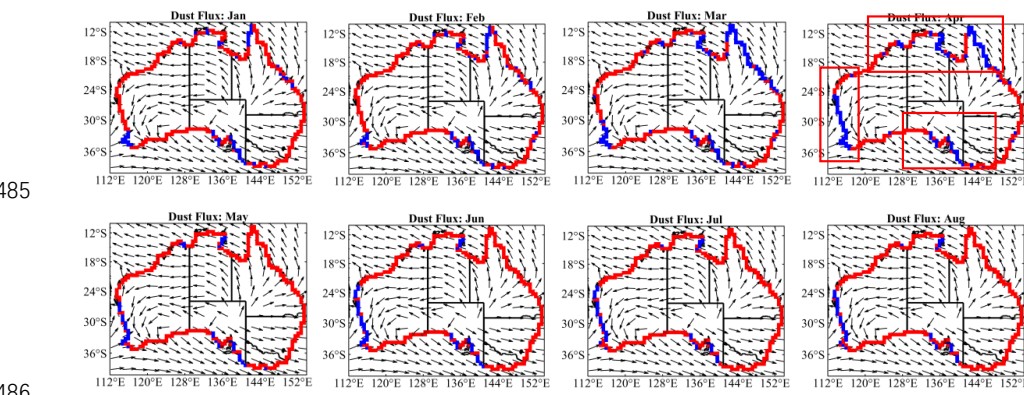





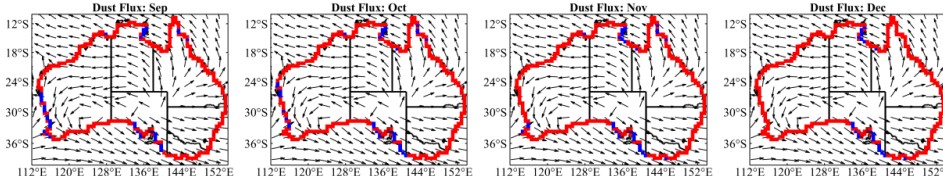

**Figure 10: Directions of dust flow in each month. Blue and red borders indicate imported and exported**

**dust, respectively.**

Figure 11 shows the annual dust budget for Australia from 1980 to 2020 using MERRA-2 aerosol reanalysis. The

green line shows the annual total dust emission, and suggests a slight decreasing trend over the past 42 years.

Overall, the Australian continent emits on average 41.84±3.10 Mt/yr into the atmosphere from 1980 to 2020, of

which 19.63±2.48 Mt/yr is exported from Australia, 1.34±0.55 Mt/yr imported from non-Australia dust sources,

and 23.56±2.99 Mt/yr deposited over the land area. The average annual dust emission over each decade from the

1980s to the 2010s are 43.12 Mt/yr, 42.63 Mt/yr, 41.36 Mt/yr, and 39.99 Mt/yr, respectively, showing an overall

decreasing trend on a decade basis. The dust emission peaks are at 48.9 Mt/yr and 48.0 Mt/yr in 1990 and 1996,

respectively. Although most of Australia was in drought during these two years, the dust storm index based on

visibility data and weather codes for these two years is around the average from 1965 to 2009 at 180 long-term

stations (O'Loingsigh et al., 2014). The DSI for these two years was around the average from 1960 to 2000 at

both inland and coastal stations (Ekström et al., 2004). Since 1996, sharp reduction of 10.2 Mt/yr occurred in dust

emission in 1997 and the change from 1998 to 2020 is relatively small (with a standard deviation of 2.27Mt/yr

excluding 2010) to that for the period from 1980 to 1997. Dust emission reached its minimum at 31.6Mt/yr in

2010. This may be strongly related to high rainfall in Australia in 2010 when the average rainfall was 687.3mm,

which is only about 17mm lower than the highest rainfall of 710.6mm in 2000 during the period of 41 years

(rainfall data can be found at http://www.bom.gov.au/web01/ncc/www/cli_chg/timeseries/rain/0112/aus/latest.txt).

The black dashed line in Figure 11 represents the net dust export that equals to the dust leaving the Australian

coastline (total net export) minus that imported from west coastal (total net import) line from 20ºS to 36ºS on an

annual basis. The blue line shows the total annual dust deposition over the continent, and was calculated using the

mass balance equation (Eq.5). The annual dust deposition shows a similar general trend to dust emission while

opposite trend to dust export from 1980 to 2020. As annual dust deposition decreasesthe dust export increases

(from 1980 to 2009). After a low value (16.2 Mt/yr and 16.8 Mt/yr) for both dust deposition and export in 2010,



the annual dust export began to decrease, while annual dust deposition started to increase from 2010 to 2020. On
a decade basis, the annual dust export reached its maximum (22.2Mt/yr) in the 2000s, while annual dust deposition
reached the minimum of 21.1Mt/yr over the same period. During this decade, dust exported from Australia was
the closest to dust imported.

$$E + I = X + D \qquad\qquad (5)$$

where $E$, $I$, $X$, and $D$ represent annual dust emission, dust import, dust export, and dust deposition, respectively. $E$
was calculated using MERRA-2 dust emission datasets for all particle bins, $I$ and $X$ were calculated using
MERRA-2 dust flux datasets. With equation 5, $D$ can be evaluated.

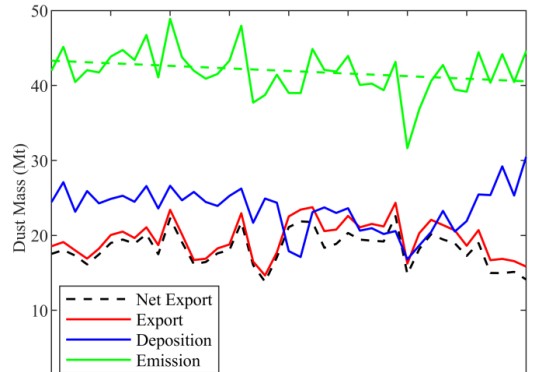


**Figure 11: Annual dust budget for Australia. Green: annual dust emission, green dashed: trend of annual**
**dust emission, red: dust export, blue: dust deposition, and black dashed: net dust export (net export-,**
**export-, deposition is the difference between emission and net export).**

Table 1 presents details of the annual dust loading for Australia in terms of clay and silt. Clay is a fine particle,
traditionally ranging from 0.1~1.0μm in radius, which corresponds to MERRA-2 dust bin001. Silt is a much
coarser particle with a broad size range of roughly 1.0μm to 25.0μm in radius. Although in the MERRA-2 dataset,
the sum of dust bin002 to bin005 only covers 1.0μm to 10μm, in this study, the sum of dust bin002 to bin005 was
regarded as silt particles. Generally, clay accounts for 6.63±0.10% of the total dust emission in Australia and silt
for 93.37±0.10%. As for dust sedimentation, almost all sediment (99.57±0.02%) is from coarse silt particles,
indicating that fine particles are more likely to be transported and exported from the Australian continent.




Table 1. Annual dust  budget for Australia in terms of clay and silt.

| | particle size (Radius) | Emission (Mt/yr) | Dry (Mt/yr) | Wet (Mt/yr) | Sedimentation (Mt/yr) |
|---|---|---|---|---|---|
| Total | 0.1~10.0μm | 41.84±3.10 | 2.65±0.35 | 4.48±0.76 | 19.76±1.17 |
| Clay | 0.1~1.0μm | 2.78±0.24 | 0.36±0.05 | 0.55±0.10 | 0.09±0.01 |
| | | 6.63±0.10% | 13.42±0.46% | 12.27±0.28% | 0.62±0.04% |
| Silt | 1.0~10.0μm | 39.07±2.86 | 2.30±0.30 | 3.93±0.67 | 19.68±1.17 |
| | | 93.37±0.10% | 86.58±0.46% | 87.73±0.28% | 99.57±0.02% |

**4.   Discussion**
There are differences in the spatial distribution of dust activity in Australia based on different indicators from
multiple data sets. Meteorological visibility-based dust event database (DEDB) and DSI were often used to
indicate the level of dust activity in Australia over the past several decades (Ekström et al., 2004; McTainsh et al.,
2011a; McTainsh and Pitblado, 1987; McTainsh et al., 1990, 1989b; McTainsh and Boughton, 1993; McTainsh et
al., 1998, 1989a, 2011b; O'Loingsigh et al., 2014). Although these indicators are quite capable of identifying the
type of dust events and the dust source areas, dust severity using meteorological observations is limited because
1)   definitions of dust events change over time, 2) only the most-coded type was recorded, 3) there is
inconsistency in records at different meteorological sites, 4) synoptic observations are subjective, 5) BoM sites
are sparse in remote areas (McTainsh et al., 2011a; McTainsh and Pitblado, 1987; O'Loingsigh et al., 2010; Strong
et al., 2011). DEDB/DSI can be a valuable reference dataset for assessing remote sensing and reanalysis products
due to the long-term coverage and distribution of BoM weather stations. The spatial distribution of dust activities
identified with DEDB/DSI differs from that based on M&M and MERRA-2, including:
1)   M&M DAOD shows that dust activities are most severe over the main dust source area, the Lake Eyre Basin,

and along major dust pathways over eastern Australia from 2002 to 2020 while the atmosphere is relatively

clean over Western Australia. DEDB/DSI (McTainsh et al., 2011a) and MERRA-2 near-surface dust

concentration show not only high dust concentration over dust source areas in central Australia but also

elevated dust concentration over downwind areas in the southeast and northwest of Australia. The main

difference between the latter two is that MERRA-2 is able to quantify dust activity with near-surface dust



555  concentrations and its variation over the Lake Eyre Basin, Nullarbor Plain, and downwind areas; on the

556  contrary, DEDB/DSI can only indicate dust activity at a few sites.

557 2) Although M&M DAOD shows high dust concentrations over eastern Australia in spring and summer, its

558  spatial pattern is dissimilar to that of DEDB/DSI. The dusty season is spring for the northern part of Australia

559  and summer for the southern part using DEDB (McTainsh et al., 1998), while two regions with high DAOD

560  were found over northern Australia in summer. In another study based on MODIS DB aerosol product

561  conducted by Ginoux et al. (2012), these two regions (the Barkly Tableland and the lee side of the Great

562  Dividing Range) were found in spring, which differs from this study but coincides with the work of McTainsh

563  et al. (1998). This is probably because 1) McTainsh et al. (1998) uses meteorological data from 1960-1987

564  while the MODIS DB data from 2003 to 2009 (Ginoux et al., 2012) and MODIS DB data from Aqua from

565  2002 to 2020 has been used in this study, 2) MODIS DB shows much higher AOD over these two regions

566  than MAIAC AOD (Shaylor et al., 2022), and MODIS DB retains high AOD for thick dust plumes (Che et

567  al., 2022). For example, for the most severe two dust storms over Australia in the last twenty years, MODIS

568  DB shows high AOD retrievals for the main dust plumes, which are even higher than 3.0, and the closest

569  AERONET AOD to satellite Aqua overpass time is much less than MODIS DB (Che et al., 2022). This

570  difference also indicates that more validation is still needed for MODIS DB aerosol products in Australia.

571 3) Another difference between M&M dataset with DEDB/DSI and MERRA-2 is that the former shows high

572  level of dust activity over the southwest of Western Australia in spring and summer while the latter two don't.

573

574 Although the MERRA-2 dust dataset is well-constrained globally and even used as reference data to validate

575 output from other models (Wu et al., 2020), the dust emission simulation for Australia in different studies vary

576 considerably (Chen et al., 2022). MERRA-2 adopts the dust emission scheme proposed by Ginoux et al. (2001),

577 which is based on soil wetness and surface wind speed. Areas of dust emission are located where rainfall is low,

578 especially below 150mm/yr (Fig.12). Discrepancy in the dust emission scheme may lead to different dust

579 simulation outputs. Wu et al. (2020) compared MERRA-2 dust emission for different regions with 15 CIMP

580 models and shows that the annual dust emission for Australia varies from 0.6 Mt/yr to 2278 Mt/yr and only three

581 out of 15 models (one with Ginoux's dust emission scheme) output similar dust emission to MERRA-2. Chen et

582 al. (2022) compared annual dust emissions in nine studies and find that the annual dust emission for Australia in

583 these studies varies within a relatively small range from 37 Mt/yr to 163 Mt/yr. Moreover, estimates of dust loading



based on ground-based data for a single dust event reveal that MERRA-2 may underestimate dust emission in
Australia. McTainsh et al. (2005) point out that published studies are very likely to overestimate dust loading
under the assumption that dust concentration is uniform from the bottom to the top and recalculated dust loadings
for dust storms on 8 February 1983 Melbourne, 20–30 May 1994 South Australia, 1 December 1987, and 23
October 2003, which were 1.23Mt, 3.3-6.4Mt, less than 3.35-4.85Mt, and 3.35-4.85Mt, respectively. These single
dust event over a region produced close or even more dust than MERRA-2 monthly total dust emission from the
entire Australia (3.58Mt, 2.35Mt, 4.76Mt, and 5.49Mt for corresponding months), indicating that MERRA-2
might underestimate dust emission in Australia.

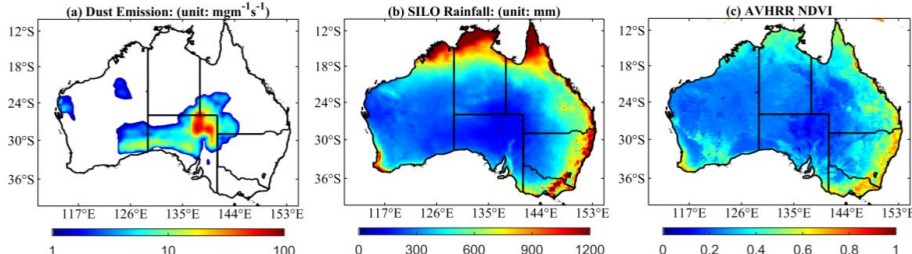


**Figure 12: Mean annual MERRA-2 dust emission (a), SILO mean annual rainfall, and (c) AVHRR NDVI**
**from 1982 to 2019.**

There is an inconsistency between dust emission and dust wet/dry deposition in Australia. This is because the rate
aerosol deposition based on MERRA-2 is be affected by its incremental update procedure and the bias in the
underlying aerosol model (Wu et al., 2020). The average dust emission (41.84Mt/yr.) plus dust import (1.34Mt/yr.)
is much higher than the averaged dust dry/wet deposition (7.13Mt/yr.) plus dust export (19.63Mt/yr.) from 1980
to 2020. The difference of approximately 16.4Mt/yr. or 38% suggests that the MERRA-2 dust deposition needs
to be adjusted to balance with dust emission. If dust sedimentation replaces dust deposition in the dust balance,
there is still a difference of 3.80Mt/yr., indicating that equation 5 still doesn't hold.

The long-term broad scale of reanalysis and remote sensing are most useful for studying climatic controls over
dust activities. In the early 1990s, Yu et al. (1992, 1993) investigated the impact of rainfall in dust source areas in
the previous autumn on dust event days in the following summer. At a much larger scale McTainsh et al. (1998)
find that climatic drivers, i.e. wind run and soil moisture, to dust storm frequencies in the north and south parts of
eastern Australia work differently in the dust seasons using BoM meteorological observation data. Although



McTainsh et al. (1998) state that the real control upon dust storms is soil moisture and associated surface
vegetation rather than rainfall, Dust emission shows a similar spatial pattern with rainfall (Fig. 12). The statement
by McTainsh et al. (1998) about climatic drivers is based on BoM meteorological records in Eastern Australia.
Their conclusion may not apply to Central Australia predominated by long-term low rainfall and sparsely
distributed vegetation. On the basis of McTainsh et al. (1998), Ekström et al. (2004) investigated the relationship
between Australian dust storms and synoptic pressure distributions and find that spring-summer dust storms over
central and southern Australia are most likely controlled by cold fronts with no precipitation and the summer-
autumn dust storms are most likely controlled by the driest period of the year. Speer (2013) finds that westerly-
induced dust storms that transport dusts to the east coast tend to occur during El Nino years and positive and
negative phases of the southern annular mode (SAM). With remote sensing products, Yu and Ginoux (2021) assess
how ENSO and the Madden-Julian Oscillation (MJO) influence dust activities in Australia. They further reveal
that during MJO phases dust activities are impacted by anomalies in convection and wind due to MJO and soil
moisture and vegetation due to ENSO.
**5.   Conclusion**
On the basis of Che et al. (2022), this paper built a DAOD dataset based on MODIS DB and MERRA-2 aerosol
datasets. Additionally, it has validated the MERRA-2 near-surface dust concentration, MERRA-2 estimated PM10,
major dust pathways, MERRA-2-MODIS combined DAOD (M&M) with collected ground-based data and with
these data sets analyzed the spatial and temporal distribution of dust activity over Australia from 1980-2020. The
M&M DAOD dataset was found to be of acceptable accuracy in Australia compared with AERONET data. M&M
DAOD contributes to long-term dust research in Australia. A power law relationship (similar to previous studies)
has been found between MERRA-2 hourly near-surface dust concentration and BoM manual horizontal visibility
at three sites in eastern Australia. Monthly MERRA-2 estimated PM10 show similar variations with AQMN
ground-based PM10 observations with an $r^2$ value from 0.14 to 0.44 at 6 selected sites; however, MERRA-2 PM10
is not sensitive to low PM10 in winter, peaks in summer and is very likely to miss extreme high monthly PM10
values. MERRA-2 flux in MERRA-2 aerosol reanalysis shows similar general dust pathways (Bowler, 1976;
Sprigg, 1982) suggesting that both early simulations and MERRA-2 are all reliable in identifying dust pathways.
Moreover, MERRA-2 further provides quantitative details on dust concentration and fluxes in a spatially
consistent manner.




Dust events over Australia are shown to be concentrated in the north and southeast in spring (Sep-Nov), occur
anywhere to the east in summer, with the dust season finishing in autumn (based on M&M DAOD). Three main
dust regions have been identified. These includethe southwest of Western Australia, and the north and south of
eastern Australia. Dust events over the southwest of Western Australia only span two months, starting in
September and reaching their peak in October. Dust events to the north of eastern Australia start in October,
gradually reaching a peak in December and January, ending in April. Dust events for the north of eastern Australia
start in November, with low levels of activity in December, reaching a peak in January and gradually end in April.

Near-surface dust concentrations were found to be the highest (over $200\mu g/m^3$) over the center of Lake Eyre Basin,
and weakened radially according to distance from the center, decreasing to below $20\mu g/m^3$ along the two main
pathways to the southwest and northeast. The dust pathway in the southeast shows lower near-surface dust
concentrations than northest, coinciding with the fact that dust entrained in central Australia is hardly transported
to the east coast (Speer, 2013). This is also shown by the ratio of near-surface dust concentration to PM10, where
high values are concentrated around central Australia and relatively low ratios (below 0.5) are found in eastern
Australia.

Total dust emission was estimated to be 40 Mt (mega-tonnes) per year over the period 1980-2020, of which nearly
50% was deposited on land; the rest as net export from the Australian continent. In the 2000s, more dust was
exported than over other periods, 22.2 Mt/yr vs. 18.7 Mt/yr, and the closest to dust deposition (21.1 Mt/yr, the
lowest); however, approximately 24 Mt/yr was deposited over the land area over other periods. Among these
particles, 6.63±0.10% (2.78±0.24Mt/yr) of emission was clay particles and almost all dust sedimentation
(99.57±0.02%) consisted of silt. This indicates that exported dust from Australia is mainly composed of fine
particles (clay). Additionally, dust import was identified from the north, south, and west coastlines using MERRA-
2 flux data. Only dust across the coastline in southwest of Western Australia was genuinely exported from other
continents while other imported dusts are sourced and recycled from exported dusts from Australia across the
north and south coastlines.

*Data availability.* The MERRA-2 and MODIS DB products are publically available from



https://search.earthdata.nasa.gov/search. The PM10 data can be downloaded from
http://www.environment.nsw.gov.au/AQMS/search.htm and the hourly horizontal visibility is available from the
Australian Boreau of Metorology. The SIlO rainfall data can be accessed from
https://www.longpaddock.qld.gov.au/silo/gridded-data/.

*Author contributions*. Yahui Che: Term, Conceptualization, Data curation, Methodology, Software, Visualization,
Investigation, Writing – original draft, preparation. Bofu Yu: Supervision, Writing – review & editing. Katherine
Parsons: Writing – review & editing."

*Conflict of interest statement*. The authors declare that they have no known competing financial interests or
personal relationships that could have appeared to influence the work reported in this paper.

*Acknowledgement*. We are thankful to the Australian Bureau of Meteorology for observing and maintaining the
horizontal data. The first author acknowledges Griffith University for the financial support provided through the
Griffith University International Postgraduate Research Scholarship (GUIPRS) and the Griffith University
Postgraduate Research Scholarship (GUPRS).



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
