# Peer review of "Temporal and spatial variations in dust activity in Australia based on remote sensing and reanalysis data sets"

_EGUsphere, 2023_

## Author Comment (AC2)

**Responses to reviewer comments**

Dear reviewer:

Many thanks for the constructive comments. We have carefully considered and addressed each comment in blue with the original comments in *italics*. Changes are highlighted in yellow in the revised manuscript.

Best regards,
Yahui Che on behalf of all authors
9-December-2023

*Reviwer: The present study analyzed and evaluated the spatial and temporal variations in dust activity in Australia for the period of 1980-2020 by using MODIS and MERRA-2 combined (M&M) DAOD, and the retrieved near-surface dust concentrations and PM10 were validated with ground-based visibility data sets and ground-based PM10 observation, respectively. With this, the authors further explored the MERRA-2 dust flux with major dust pathways identified in previous studies and quantified the annual dust cycle for Australia over the period from 1980 to 2020, that providing useful information to evaluate the adverse impacts stemming from dust events in Australia, although some similar attempts had been conducted before. The manuscript was well written, Still, there are some things are unclear/invalid, and more discussion should be added to make the analysis more complete. Therefore I recommend the publication of this study after some issues were properly revised and improved.*

We appreciate this positive comment, and we have rewritten or revised the contents in response to the reviewer's comments.

*Specific and technical comments:*

*1. Abstract, the authors claimed that the MERRA-2 dust flux shows the same major dust pathways as those in previous studies and similar dust emissions/deposition areas. What about the source of those previous studies? From the ground observation data or from the MODIS products?*

It is based on the ground-based observation data. The same major dust pathways in previous studies were identified using the orientation of sand dunes during the active dune building period (Bowler, 1976: https://doi.org/10.1016/0012-8252(76)90008-8, Sprigg 1982, Blewett, 2012: http://epress.anu.edu.au).

"Moreover, MERRA-2 dust flux shows the same major dust pathways as those in previous studies and similar dust emissions/deposition areas identified using ground-based observations." (Lines 16-17, Page 1)

*2. Line 77, the limitations are referred to the study of Yang et al., (2021) or from the previous related studies? If it was the former, it is better to summarize the limitations from previous studies. In addition, I note that the author had published a similar study related to the MERRA-2 AOD and DAOD with MODIS which also focus on the dust intensity in Australia (Che et al., 2022), what about the difference with the present study?*

The limitations are referred to in the studies of dust intensities with MODIS (Yu and Ginoux, 2021) and MERRA-2 data (Yang et al., 2020).

Seasonal DAOD over Australia can be found in Yu and Ginoux (2021) with MODIS and in Yang et al. (2020) using MERRA-2 data. However, validation of MERRA-2 DAOD had not been reported in previous studies and we don't know which DAOD is more appropriate for dust research in Australia. Che et al. (2022) aims to investigate the performance of MERRA-2 AOD and DAOD using MODIS and AERONET data. Seasonal AOD over Australia from 2002 to 2020 is used in Che et al. (2022) to find out the difference between MERRA-2 and MODIS in terms of AOD and DAOD. "The limitations" we conclude in line 77 are sourced from Che et al (2022).

For a clearer statement, the sentence

"There are limitations to this kind of study."

has been replaced with

"There are limitations to this type of investigation using MODIS and MERRA-2 data." (Line 80, Page 3)

*3. Line 90-93, it is unclear about the limitation of using site-based PM10 observations here with the example of de Jesus et al. (2020).*

We have added the limitation of de Jesus et al. (2020) in the revised manuscript.

"For example, de Jesus et al. (2020) analyzed PM10 trends in major cities of Australia over the last two decades using site PM10 observations, while they did not conduct a trend analysis specifically for dust concentrations because the dust component could not be accurately obtained from PM10 which consists of both dust and non-dust particles from multiple sources." (Lines 94-98, Page 4)

*4. Line 163-165, 11). Are both the hourly BoM horizontal visibility data and the dust type SYNOP code were recorded by a weather observer or not? In addition, the manually estimated visibility have an upper limit of 10km, some low dust or PM10 condition would be unproperly analyzed with visibility data, did the authors compared the manually dataset with data from the automated instruments? like the Atmospheric Visibility Sensor which could be used in recent years.*

Not every observation of horizontal visibility was assigned with a weather type.

We agree with the reviewer that the inclusion of visibilities of 10km would result in improper validation, so visibilities of10km have been removed in Fig.6.

[Figure]

Figure 1. The updated figure 6 in the revised manuscript.

Also, we have replaced:

"In this study, all hourly BoM horizontal visibility data with a dust type SYNOP code 164 (excluding thunderstorms) were used for validating MERRA-2 near-surface dust mass concentration at three sites 165 (Charleville, Cobar, and Longreach, Fig.1) from 1980 to 2020."

With

"Manual observations of dust activities (using horizontal visibility as a measure), span from the early 19th century to present (O'Loingsigh et al. 2017). In this study, to ensure the consistency of data points to dust storm research, hourly BoM visibility observations of less than 10km with a dust SYNOP code (excluding thunderstorms with raised dust) were used. These observations were used to validate MERRA-2 near-surface dust mass concentration at three sites: Charleville, Cobar, and Longreach (Figure. 1) from 1980 to 2020." (Lines 185-189, Page 8)

Figure 2 shows the comparison of BoM horizontal visibility and AWS visibility at Longreach . An r2 value of almost 0 does not indicate significant correlation between the two. Overall, the difference between the two increases with the increase of BoM horizontal visibility. The large difference between the two for the visibility greater than 2km would lead to different conclusions about MERRA-2 near-surface dust concentrations depending on which visibility data set is used. Considering a large number of dust studies in Australia were based on with horizontal visibility data, horizontal visibility data were used to be comparable to previous studies.

[Figure]

Figure 2. Comparison of BoM horizontal visibility and AWS visibility at Longreach station. The blue line represents the 1:1 line.

*5. Line 189-206, this part should be moved to the Introduction.*

This part has been moved to the introduction in the revised manuscript. (Lines 125-134, Page 5)

*6. Line 224-225, the Eq 3 used in this study is not considered the contribution of nitrate aerosol, evidence about the minor role of nitrate aerosol for PM10 needed to be provided.*

Some evidence has been added to the revised manuscript.

"Considering that coarse mode aerosols take up 57%-71% of total aerosols over major cities (Chan et al., 2008) and nitrate emissions contribute much less than other aerosol species to the atmosphere (Bauer et al., 2007), especially during smoke events in the northern savanna (Desservettaz et al., 2017), in this study, the method developed by the Global Modeling and Assimilation Office (GMAO) (equation 7) using MERRA-2 3-D aerosol mass mixing ratios was used for PM10 estimation over Australia. Nitrate is missing due to its minor contribution to PM10 concentrations." (Lines 249-254, Page 10)

*7. Line 366-368, it is unclear why using the example of Shao et al., (2017), it seems not support the conclusion below. Please added more explanation to clarify it.*

This sentence has been moved to the discussion in the revised manuscript.

*8. Line 497-501, previous studies based on visibility data, weather codes and DSI seems not support the dust emission peaked in 1990 and 1996, please explain it.*

The reason why DSI records don't support the MERRA-2 dust emission has been added in the revised manuscript.

"The discrepancy of this and from the previous studies is caused by a lack of BoM weather observations in central Australia where dust emission is concentrated (Fig. 12)." (Lines 534-536, Page 24)

*9. Line 598-602, what the possible reasons caused the imbalance between source and sink of dust? In addition, the data source or the calculated of averaged dust dry/wet deposition, and the dust sedimentation should be provided here.*

The discussion about dust dry/wet deposition, and dust sedimentation is based on MERRA-2 product. This has been indicated in the revised manuscript.

"There is an inconsistency between dust emission and dust wet/dry deposition in Australia using MEERRA-2 product."

The possible reason why there is imbalance between dust emission and deposition is that:

"The imbalance between MERRA-2 dust emission and deposition could be caused by the incremental update procedure in MERRA-2 and a lack of data assimilation for non-AOD parameters (Wu et al., 2020). Only AOD in MERRA-2 was constrained by AOD observations from multiple sources (Randles et al., 2017), leading to the bias in the dust component in the underlying aerosol model and in biased dust emission data that could not be corrected." (Lines 682-686, Page 30)

*10. Line 604-621, Discussion in this part is weak and meaningless, as the main finding of the present study is not included. Is it consistent or deepen our understanding on the climatic control over dust activities compared with the previous studies?*

This paragraph has been rewritten in the revised manuscript:

"The long-term broad scale of dust activities in this study would provide useful information on dust activities in Australia. Most dust studies in Australia are based on ground-based observations. In the early 1990s, Yu et al. (1992, 1993) investigated the impact of rainfall in dust source areas in the previous autumn on dust event days in Mildura, Australia in the following summer. At a much larger scale, McTainsh et al. (1998) find that climatic drivers, i.e. wind run and soil moisture, affect dust storm frequencies in the north and south parts of eastern Australia in different ways during the dust season based on BoM meteorological observations. On the basis of McTainsh et al. (1998), Ekström et al. (2004) investigated the relationship between Australian dust storms and synoptic pressure distributions and find that spring-summer dust storms over central and southern Australia are most likely controlled by cold fronts with no precipitation and the summer-autumn dust storms are most likely controlled by the driest period of the year. Speer (2013) finds that westerly-induced dust storms that transport dust to the east coast tend to occur during El Nino years and positive and negative phases of the southern annular mode (SAM). Compared to ground-based observations, satellite remote sensing data provide dust related parameters at a broader scale. For example, with remote sensing products, Yu and Ginoux (2021) assess how ENSO and the Madden-Julian Oscillation (MJO) influence dust

activities in Australia. They further show that during MJO phases dust activities are impacted by anomalies in convection and wind due to MJO and soil moisture and vegetation due to ENSO. This study provides the spatial pattern of dust activities in Australia using MERRA-MODIS combined DAOD and MERRA-2 PM10 and near-surface dust concentration. The findings in this study serve as an extension of previous studies to deepen our understanding of the spatial pattern of dust activities in Australia, and provide useful information on dust activities for future dust research in Australia." (Lines 691-709, Page 31)

---

## Author Comment (AC3)

**Responses to reviewer comments**

Dear reviewer:

Many thanks for the constructive comments. We have carefully considered and addressed each comment in blue with the original comments in *italics*. Changes are highlighted in yellow in the revised manuscript.

Best regards,
Yahui Che on behalf of all authors
9-December-2023

*To reveal the temporal and spatial features of dust activity in Australia, the authors developed a new product of MODIS and MERRA-2 combined (M&M) DAOD. They further evaluated tis MODIS-MERRA DAOD by using multiple measurements. Then the authors presented many results on the spatial variations and seasonal features in dust activity.*

*Overall, this is a well-conducted study and the authors show some interesting results. The developed combined DAOD could be very helpful. However, I believe this manuscript can be improved by doing more in-depth analysis on the long-term trend of dust activity. Please find my comments below.*

Thanks for the positive comment. The comments below have been addressed and revisions have been made in the revised manuscript.

*My major concern is about the analysis on temporal variations of Australia dust activity. The authors mainly talked about long-term trends of dust budget from 1980 to 2020, but in fact a further discussion on the key factors driving these trends is needed. In addition, Fig.12 only shows mean dust emission over 1982-2019; I think a spatial trend for dust emission should be given which might be linked to variations in rainfall and land cover.*

A spatial trend analysis has been added:

"As dust emissions are primarily determined by soil moisture content and surface wind speed (Ginoux et al., 2001), rainfall is strongly correlated with dust emissions (Figures 12a and b). Areas of dust emission, such as the center of the Lake Eyre Basin and the Nullarbor Plain, are typically associated with low rainfall, especially below 150mm/yr. The Mann-Kendall (MK) tests conducted on annual dust emission and rainfall data from 1980 to 2020 further highlight the substantial inhibitory impact of rainfall on dust emissions. A decreasing trend of dust emission occurred over the past 40 years with increased rainfall for almost all regions, especially in southwest of WA ((figure 12e). Northern Australia commonly shows an increasing trend of rainfall, while dust emissions remained essentially unchanged. This is because the highly vegetated surface in Northern Australia rarely emits dust particles. Conversely, with significant decreasing rainfall (p < 0.05), the southwest of QLD, as a part of the Lake Eyre Basin, shows an

increasing trend of dust emissions. Nevertheless, the impact of photosynthetically active vegetation on dust emissions was ignored in Ginoux's dust emission scheme (figure 12f), potentially resulting in uncertainties in dust emission estimates. Despite sharing a similar spatial pattern with rainfall trends, NDVI trends show an opposite trend to dust emissions in most of the Lake Eyre Basin. This indicates that the decreasing trend of photosynthetically active vegetation cover also contributes to the increasing trend of dust emissions in the southwest of QLD. It is essential to acknowledge and consider this factor in dust emission estimations.

[Figure]

**Figure 12: Mean annual MERRA-2 dust emission (a), SILO mean annual rainfall (b), and (c) AVHRR NDVI, and (d) Mann-Kendall (MK) test for annual dust emission, (e) for annual rainfall (f) for annual NDVI from 1982 to 2019. Positives and negatives represent an increasing and decreasing trend, respectively. A p-value < 0.05 for MK test is shown with green lines for dust emission (d) and rainfall (e) and gray colors for NDVI in (g)."** (Lines 636-658, Pages 28-29).

*Fig.3 looks like that the MODIS-MERRA DAOD is not in a good agreement with AERONET AOD. So I am not convinced by this validation.*

The EE (expected envelope) lines are often used to indicate agreement between satellite AOD and ground-based data (Chu et al., 2002; https://doi.org/10.1029/2001GL013205). The EE for the M&M DAOD is only $\pm(0.016 + 0.15\tau_{Aero})$, which is much lower than that for MODIS DB or MERRA-2 DAOD ($\pm(0.016 + 0.15\tau_{Aero})$ ) in Australia. Another example is that the EE for MODIS DB data on a global scale is $\pm(0.05 + 0.20)$ while they have a strong agreement, with an r value of 0.9 (Sayer et al., 2019; https://doi.org/10.1029/2018JD029598). These different EEs are not contradictory, because AOD ranges from 0.0 to more than 3 on a global scale while the maximum DAOD in this validation is around 0.1 for Australia. As a result, under low aerosol conditions, satellite data tend to have a small absolute error, with weaker correlation; however, it does not mean their agreement is not good.

*L306: what's the meaning of expected envelope and how is it determined?*

The EE (expected envelope) lines, expressed as $\pm(a + b\tau_A)$, are typically used to indicate the accuracy of MODIS AOD datasets where a and b were determined to have these two lines contain 68% of data points (Levy et al., 2010; doi:10.5194/acp-10-10399-2010 ).

Its definition has been added:

"These studies show that there is a high probability of data points (MODIS DB and AERONET) within the expected envelope (EE) lines, which are defined with two lines ($\pm(0.03+0.15\tau)$) containing approximately 68% of data points (Che et al., 2022)." (Lines 277-278, Page 11)

*L463: please replace "PM" by "PM10"*

"PM" has been replaced with "PM10"

"Figure 9: Seasonal MERRA-2 near-surface dust concentration in (a) Dec-Feb, (d) Mar-May, (g) Jun-Jul, and (j) Sep-Nov from 2002-2020; Seasonal MERRA-2 PM10 in (b) Dec-Feb, (e) Mar-May, (h) Jun-Jul, and (k) Sep-Nov from 2002-2020; and the ratio of MERRA-2 near-surface concentration to MERRA-2 PM10 in (c) Dec-Feb, (f) Mar-May, (i) Jun-Jul, and (l) Sep-Nov from 2002-2020." (Lines 492, Page 22)

*L513: what's the reason of an increasing dust deposition after 2010? How about the trend in different seasons?*

A possible reason for the increasing trend of dust deposition has been added:

"The post-2010 trend of dust deposition is likely caused by surface wind speed. Annual dust deposition shows a strong correlation with the silt particle fraction (figure 13a) for the period from 1980-2010 (blue) and post-2010 (green). Large particles have a higher settling velocity. As more than 86% of dust was deposited on the land, and dust emission and import remained largely unchanged since 2000. A higher fraction of silt particles indicates that deposited material is made up of mostly silt particles. Nevertheless, as the surface wind speed is a key factor in dust emission estimation, an increasing surface wind speed would entrain more coarse dust particles, leading to more dust deposition on the land, as indicated by an $r^2$ value of 0.44 between dust deposition with surface wind speed (figure 13b). Additionally,

Fig. 14 shows seasonal variations in the amount of dust deposition in mainland Australia with the maximum deposition occurring in summer (Dec-Feb), and the minimum in autumn (Sep-Nov). It is clear from Fig. 14 that the minimum deposition has increased from less than 1 Mt in 2011 to more than 3 Mt over the past several years (2017-2020), while the peak deposition has not changed significantly over the 11 years (Fig. 14).

[Figure]

**Figure 13: Relationships between annual dust deposition and (a) silt particle fraction, and (b) surface wind speed 1980-2016. Green points represent data since 2010; Indicators in each panel are calculated using all data points.**

[Figure]

**Figure 14: Seasonal dust deposition on mainland Australia since 2000**

" (Lines 661-676, Page 29-30)